# Potential Transformation of Contaminated Areas into Public Parks: Evidence from São Paulo, Brazil

**DOI:** 10.3390/ijerph191911933

**Published:** 2022-09-21

**Authors:** Camila Vitorino dos Santos, Helena Ribeiro

**Affiliations:** Faculdade de Saúde Pública, Universidade de São Paulo, Sao Paulo 01246-904, Brazil

**Keywords:** contaminated areas, urban residues, urban areas, environmental health

## Abstract

Waste-contaminated areas have been reused and requalified environmentally across the globe, aiming to reintegrate them into urban dynamics with new functions such as public parks. This practice has attracted the attention of public health and environmental control agencies due to the scarcity of free areas and vacant spaces for creation of green areas, and d the need for more sustainable planning guidelines in large cities. The present work aimed to study processes of requalification of waste-contaminated areas for transformation in parks, using as study two cases located in the city of São Paulo, Brazil. Method: Documentary research in environmental agencies, literature review and field visit. Results: In the two cases the process of requalification was unequal, with emphasis on three aspects: the actors involved in the case, the role of civil society and the action of the Public Prosecutor’s Office of the State of São Paulo. Thus, it becomes evident that successful cases of waste-contaminated areas in the city of São Paulo are linked to the direct support of these aspects complemented with the private sector. Furthermore, a consensus is necessary among the public authorities regarding the laws of contaminated areas versus environmental damage full repair in the process of requalification.

## 1. Introduction

In the last thirty years, as ecological awareness grows around the world, there has been a need for the implementation of more sustainable planning guidelines in large cities. With the disordered urban expansion in the territory, there is a notable scarcity of free land and vacant spaces for the creation of new vegetated areas in cities.

The insertion of a green area in urban space depends on urban and economic factors, which sometimes are unfeasible. The reuse of waste-contaminated areas, transforming them into public parks is a particularly well-executed solution performed in many countries.

A contaminated area is one in which there is evidence of contaminants on the ground or site, in quantities or concentrations above the environmentally acceptable reference values and may cause harm to human or ecological health or other goods to be protected [1].

Contaminated areas degrade the urban landscape and interfere in the quality of life of the population, since they alter the natural characteristics of the compartments and go unnoticed by the population. In addition, they are considered polluting sources of great magnitude and importance for public health, due to the risks to human health and public safety, and their complexity of management [2].

The measures of rehabilitation are based on the use of the area that may still be contaminated, with the guarantee that there are no transport routes of contaminants or receptors exposed to them. Remediation measures refer to those aimed at containing and/or isolating and/or treating contaminated environments, seeking to eliminate or reduce contaminant levels to acceptable concentrations according to legislation, allowing the area to be rehabilitated later [1].

Worldwide, there are several accidents, leaks, disposals and improper handling of materials and waste that have resulted in tens of thousands of contaminated sites. In the United States, it is estimated that there are more than 450,000 contaminated areas that require environmental intervention [3] and more than 340,000 contaminated sites in the European Union, posing risks to health and the environment and preventing the revitalization of urban neighborhoods [4].

In Brazil, despite the advance of technology and the constant search for compliance with environmental legislation, only three Brazilian states (São Paulo, Rio de Janeiro, and Minas Gerais) have data and information provided by environmental agencies on contaminated areas and their main characteristics.

According to the survey by the Environmental Company of the State of São Paulo—CETESB, on 6 December 2020, 571 contaminated areas were recorded in the state of São Paulo, Brazil. In the city of São Paulo, the largest Brazilian metropolis, there were 1103 areas that had undergone the rehabilitation process and were able to be declared rehabilitated, indicating a tendency to change land use and occupation [5].

Solid waste was the fourth source of contamination, with 208 contaminated areas registered in the state of São Paulo. For the safe reuse of these areas, remediation measures should be carried out to recover them, or to make current use compatible with the future [5].

In the State of Minas Gerais, Brazil, an inventory carried out by the State Environmental Foundation (FEAM), in December 2021, indicated 687 contaminated and rehabilitated areas, 20 of which were managed by the City Hall of Belo Horizonte. As in the state of São Paulo, the largest number of enterprises with contaminated sites corresponds to the activity of fuel stations, including the retail trade in fuels and dealers of gasoline, alcohol, and diesel [6].

Geographically, the metropolitan region of Belo Horizonte is the one with the highest number of areas in the list of contaminated and rehabilitated areas. Regarding areas rehabilitated for declared use, there was an increase in compliance with legal requirements, from 224 areas in 2020 to 251 areas in 2021, indicating an improvement in the management of contaminated areas [6]. Municipal solid waste was the second polluting source, due to irregular disposal points on public roads and many clandestine dumps [6].

In the state of Rio de Janeiro, according to the Institute of the Environment (INEA), the Register of Contaminated and Rehabilitated Areas identified, in 2019, 327 contaminated areas in 48 municipalities. Sites were contaminated by gas stations, industries, and waste landfill. In that year, only nine areas were rehabilitated for declared use, i.e., for new uses and functions [7].

The dumping of solid waste in the open varies significantly between different regions of the world since it is related to income and standard of living of cities [8].

In the richest countries that generate larger amounts of waste, there is more capacity to equate management, by a sum of factors that include economic resources, environmental concern of the population and technological development.

Globally, in cities of low- and middle-income countries, due to the very accelerated urbanization, there are deficits in the financial and administrative capacity for providing essential infrastructure and services such as water, sanitation, collection and proper disposal of garbage, housing, and in ensuring safety and control of environmental quality for the population. Thus, dumps continue to be the main method of disposal of municipal solid waste, significantly impacting the environment and public health [8,9].

A rehabilitated area for declared use is one that, after being submitted to intervention measures, presents a level of tolerable risk for future use, considering human health. Reusing a contaminated area means reinserting it back into the urban fabric, making it an instrument of urban requalification [10].

The city of São Paulo has many areas subject to the management of contaminated sites compared to others, especially due to the environmental licensing program conducted by CETESB, required by the federal government since 2000 [11].

The creation of green areas and the requalification of public spaces are relevant to public health and the environment. They are considered as one of the essential items for the well-being of the population living in large cities and a great tool to plan and develop a more sustainable urban environment [12].

Among these benefits are the conservation and reintroduction of species of native fauna and flora, the improvement of air and water quality, the climate balance and consequently thermal comfort. Due to this, they are used as indicators of quality of life and collective satisfaction, being directly linked to leisure and recreation activities through social interaction [12,13,14].

In addition, green spaces are associated with psychological well-being, decreased symptoms of depression, anxiety, and stress, providing a higher quality of life to the population [15,16]. With this, to alleviate urban problems stemming from urban expansion and the difficulty of creating public parks, managers have been rethinking the potential for reuse of contaminated areas as a sustainable tool for land use and occupation.

Considering this scenario, the present study aimed to analyze two case studies of requalification of urban solid waste-contaminated areas for transformation into parks, in the city of São Paulo, Brazil.

## 2. Materials and Methods

For the present study, the methods adopted were case studies, using documentary and bibliographic research and field visit. A case study of two parks located in rehabilitated areas in the city of São Paulo, Brazil was carried out. The municipality of São Paulo has 11,811,516 inhabitants residing in its territory of 1521 km^2^ and a population density of 7765 inhabitants/km^2^, being the main one in the metropolitan São Paulo area. It concentrates all the activities of economic, cultural, and social interests of the city [17].

Two areas contaminated by solid waste located within the municipality were selected, with declared use for the implementation of a public park.

The selection criteria were based on the following aspects: area insufficiently studied and explored; time in the scheduling of environmental studies at the Municipal Bureau of Green and the Environment (Secretaria Municipal do Verde e do Meio Ambiente de São Paulo—SVMA); time for the implementation and completion of the park’s construction; the potential for reuse of the area for new uses, and the benefits that the park might bring to the surrounding community and future users.

The first case selected was the contaminated area of the Jardim Primavera Municipal Park (Figure 1), inserted at the eastern part of the municipality, in the district of Vila Jacuí, in São Miguel Paulista.

The second case was the contaminated area of the Villa-Lobos State Park (Figure 2) already fully implemented in the city of São Paulo. The Villa-Lobos State Park is in the neighborhood of Alto de Pinheiros, in the west, in Pinheiros district.

The selection criteria were similar characteristics such as land use and occupation and source of pollution; operation period and deactivation of the polluting source; contaminated compartments (soil, surface, and groundwater); and considered as a successful case of requalification of waste-contaminated area and the current or future use of the area.

The documentary research was carried out after telephone appointments at the public institutions Municipal Bureau of Green and the Environment (Secretaria Municipal do Verde e do Meio Ambiente de São Paulo—SVMA) and the Environmental Company of the State of São Paulo (Companhia Ambiental do Estado de São Paulo—CETESB) to consult the administrative processes and environmental studies on the sites of the parks.

The survey of secondary data on the contaminated area of Jardim Primavera municipal park was carried out from February to December 2018. This survey included data on the background of land use and occupation; type of contamination; impacted means; contaminants; environmental management steps performed; actors involved; intervention measures taken; implementation of the park and current environmental situation of the area.

Subsequently, from May to June 2019, an appointment was made at the Department of Infrastructure and Environment of the State of São Paulo (Secretaria de Infraestrutura e Meio Ambiente do Estado de São Paulo—SIMA), Urban Park Coordination (Coordenadoria de Parque Urbanos—CPU) to carry out consultations with environmental and technical documents about the Villa-Lobos State Park. This consultation aimed to obtain the same detailed information as for the Jardim Primavera Municipal Park. The visits and photographic records were carried out concomitantly with the documentary research.

### 2.1. Background: Soil Use and Occupation

#### 2.1.1. Jardim Primavera Municipal Park

The area occupied by the Jardim Primavera Municipal Park until the 1950s presented a small earthmoving operation, but around 1968 the exploration of sand by a private company began, a common activity in various regions of the city. This activity resulted in the formation of two pits, one larger and the other smaller, practically occupying the entire property, representing great environmental damage (Figure 3). Due to the interception of the groundwater, ponds were formed in these pits [20].

The mining pits had an average depth of 25 m in relation to the natural level of the terrain, reaching up to 40 m. The larger pit had an area of 60,784 m^2^ and the smaller one measured 11,750 m^2^, and due to this depth and rainfall at the site, the area began to present risks of drowning for the population living in the surroundings, Figure 3 [18,20].

Around 1979, the city of São Paulo was pressured by the local population to ground the pits that began to receive urban solid waste at the site. In an immediate response to the population, the smaller pit was grounded with soil from the leveling of the land, and the larger pit continued to receive construction residues (rubble), but there was no control for its disposal. The site also received other types of waste from 1979 to 1988 [18,21]. Urban solid waste was arranged randomly on the larger mining pit in precarious operating conditions and without technical measures for soil and groundwater protection, turning it into a dump [22].

In mid-1981, with the complaint of the population to the government (Regional Administration of São Miguel Paulista, currently the subprefecture) because of the precarious conditions of the area and the need for another place for the disposal of garbage in the east, the City Hall of São Paulo proposed to build a landfill site. Thus, the Jacuí landfill project was elaborated in October 1983, which had a favorable technical report by CETESB [23].

The Jacuí landfill was designed in such a way that it would consist of two smaller landfills, to take advantage of the two pits. However, due to the urgency of a place for the disposal of waste, gas drains were installed in the pit area, which collected and burned the gases generated by the 2.5 tons of waste deposited, until operation ended in 1988 [20].

Also in 1988, near the end of the landfill’s useful life, CETESB was triggered by the residents of the neighborhood to verify the migration of gases from the landfill to some surrounding residences. This migration caused an explosion accident in a surrounding house, with two people injured, due to the presence of biogas in a nearby well [23].

From then on, between 1988 and 1994, in successive technical inspections of CETESB, it was verified by means of measures with explosimeters the presence of biogas in several places of neighboring residences, such as at the exit point of the conduit of installation of a doorbell, on the sidewalk, in a water well, in stormwater drainage drains, in existing cracks in the ground and in the paving of the streets. As a result, CETESB determined the closure of all water supply wells in the region, retaining only one for monitoring, and requested the construction of deeper drains in the area [23].

In parallel, in 1990, the municipality developed the pilot project of the park called Primavera. The project was elaborated through the Department of Parks and Green Areas (Departamento de Parques e Áreas Verdes—DEPAVE) linked at the time to the Municipal Bureau of Services and Works (Secretaria Municipal de Serviços e Obras—SSO), without previous studies to evaluate soil and groundwater contamination [22].

However, the project already raised concerns about the possible risks to human health that the area presented. Few buildings and sports courts with lawns were planned to be built outside the area grounded with solid waste, avoiding direct contact with soil and confinement of gases. In addition, afforestation should consider planting species of surface roots in the grounded area, and deeper roots in places that maintained natural soil [22,23].

Without previous studies, the execution of the project began, which was then paralyzed due to the detection of gases during the implantation, making it impossible to use the land for the intended purpose. Among the identified problems there were strong odor of gas, bubbles caused by fumes, bubbles in the puddles of water including external to the landfill massif and the difficulty for planting the lawns. These problems caused the work to stop in 1991, with the suggestion of waiting for the stabilization of the terrain and elimination of gases [23].

#### 2.1.2. Villa—Lobos State Park

Villa-Lobos State Park is already finished and in use. It belongs to the government of the state of São Paulo. In the area of the park, located in an old floodplain of the Pinheiros River, the occupation began in 1958, where it presented grassy vegetation, some paths of earthmoving and the Boaçava stream (Figure 4).

In its surroundings, there were allotments and residential occupation, in addition to some developments in the process of construction, such as the Olympic Rowing Lane of the University of São Paulo—USP, empty land and two industrial sheds. In 1968, the area had more dense stretches of vegetation, and in its southern portion, near the Pinheiros River, a great earthmoving with ditches and roads used for the extraction of sand [19].

The tailings formed in the area ranged from 5 to 11 m high, in the form of piles in the northern part. There were also small ponds in the western part (Figure 5). Its surroundings, to the north, had denser residential occupation and allotments, to the west the expansion of industrial occupation, and to the south, bordering the Pinheiros River, the implementation of CPTM Railway Road-CPTM, the current Esmeralda line [19].

Later, in 1974, in the southern stretch of the area, vegetation was recomposed in a pit, and in the northeast portion, a new area of earthmoving was made. In 1977, the pit was completely grounded, in which it was also possible to notice two sediment dredging ferries of the Pinheiros River in front of the park area, and signs of dredged material released in a lagoon formed in the central part, next to the old sand extraction pit.

As for its surroundings, the north and east were completely occupied by residences, the University City of the University of São Paulo was installed to the south, as well as its Olympic Rowing Lane, and to the west there was the increase in industrial activity. It was possible to observe the Avenida Marginal of Pinheiros river already implanted. In 1986, new paths and decanting dikes of dredged material from the river were formed within the area. In the central area, a soil landfill of high proportions, apparently without construction debris, is identified and in the northwest a stretch with residential occupation and a soccer field [19].

During this period, near the site where the administration of the park operates nowadays, an area was identified that received metal materials for recycling, apparently scrap from underground storage tanks. The surroundings at that time had the same characteristics as in 1968, but with increased occupation (Figure 6).

Still in 1986, the area was used in its westernmost portion as a garbage dump of the Entrepots and General Warehouses Company of the State of São Paulo (Companhia Entrepostos e Armazéns Gerais do Estado de São Paulo—CEAGESP), from where about eighty families collected food and packaging. In the eastern part, adjacent to the current Villa-Lobos Shopping Mall, dredged material from the Pinheiros River was deposited and in the central portion, the former owner allowed the deposit of rubble, derived from construction sites.

In the early 1980s, the architect Decio Tozzi, on his way back and forth from FAU-USP, passed daily through this immense and degraded area, which was one of the last urban voids of the expanded center of São Paulo. The architect had the idea of recovering the area, transforming this open-air dump into a park, for which he elaborated a project. A contemporary park was designed, with large and dense biodiverse forest, composed of 50,000 trees divided into 300 species and 12 grassy clearings intended for free use of the population [24].

In the center, an area was designed by rows of imperial palm trees where an urban landscaping was developed, composed of a succession of small squares with shade trees, aiming to represent the landscaping of the squares and urban gardens of Brazilian cities. Continuing, the architect presented the project to three councilors who liked the idea. Later, for two and a half years, he sought to arouse the interest of the population, disseminating the project in neighborhood associations, community chambers, schools, clubs and even in communities. Once popular support was gained, the idea was brought to the state government, which took charge of expropriating the land.

In 1987, the year of commemoration of the centenary of the birth of composer and musician Heitor Villa-Lobos, the first studies were presented, aiming at the implementation of the contemporary park, which had music as the main theme [24]. The executive project was completed by the architect and the construction was put in charge of the Camargo Correa construction company, the winner of the bid. Existing buildings in the park area were removed in 1988.

In 1989, the Villa-Lobos State Park began to be deployed, starting with earthworks, Figure 7.

The entire area of the park received a clean soil cover with known origin, in variable layers to level the existing elevations, preventing direct contact with the deposited material and the flow of rainwater (Figure 7). The Boaçava stream, which passed through the area, was channeled. All the residue was kept in the ground. After earthmoving, the park was 6 m taller than the avenue next to it. In this period, families living on the site were removed, 500,000 m^3^ of rubble more than 1 m in diameter was removed, and 2 million m^3^ of rubble was moved.

The next step was to convert the degraded and poorly permeable surface into fertile soil. To this end, a rigorous program of organic fertilization was developed using mainly vermicompost. In the space between the pits of the trees the green fertilization—the planting of legumes of annual cycle to incorporate nitrogen and organic matter into the soil—was also made. However, before the fruits and seeds appear, the green mass was cut. Finally, the soil underwent correction with application of limestone and chemical fertilizers based on NPK.

In the early 1990s, the trees began to be planted and organize the formation of the forests, the land was divided into modules of 10 m × 10 m and each of them received 25 seedlings with an interval of 2 m. The planted species are distinct among native pioneers (Angico, Candeia, Embaúba), non-pioneer natives (Cedar, Fig tree, Jatobá), species of marked flowering (Ipê-amarelo, Ipê-branco, Ipê-roxo) and species that attract birds (Mulberry, Guava, Jerivá palm tree), totaling 50,000 trees and 300 different species [24].

In 1994, the park was inaugurated unfinished, not in the original size proposed, with several sound installations, exhibition buildings and schools of ballet and music. The eastern area was incorporated after ten years. In 2006, the park was delivered completed with approximately 24,000 trees planted in pits of 1000 L of substrate and soil exchange. Two-sport and tennis courts, Cooper track, skater track, amphitheater, bike lanes, soccer grounds and parking with 730 spaces [19] were made.

In that same year of 2006, civil society composed of environmental entities opened a Public Civil Action against the state, with the participation of the Public Prosecutor’s Office, which ended in the form of an agreement. The action questioned many points of the park as: the delay of the opening of the entire area of the Villa-Lobos State Park to the population; the priority in the implementation of forests, lawn spaces, pedestrian paths; the interest of the population in expanding the spaces of culture, leisure, sport and green areas available; the need to meet and have equipment for people with locomotion disabilities; the restriction for the practice of large sporting or musical events; among others [25]. Faced with these issues, an agreement was established between civil society and state stakeholders with the participation of the Public Prosecutor’s Office, clearly stating the attributions and responsibilities of each one regarding the operation and conservation of the Villa-Lobos State Park.

In 2008, 800 more seedlings were planted for Autoban’s Environmental Recovery Commitment Term (TCRA) for forest enrichment. In 2009, the Villa Ambiental space was inaugurated, the new headquarters of the park’s administration and a headquarters of the 1st Company of the 23rd Battalion of the Military Police were also opened in the park. In 2010, the Ouvillas space and the Ruth Cardoso Orchid Greenhouse were inaugurated, and in 2013, the environmental education center [19].

## 3. Results

The results found in the contaminated areas of the respective parks are presented. In both cases, contamination was derived from the irregular disposal of urban waste directly in the soil, which was submitted to the management of contaminated areas established by CETESB for the process of reuse and rehabilitation, in the form of a public park.

The practices adopted in the requalification process presented common and different aspects, although both areas are located within the city of São Paulo.

The main common aspects identified areas related to legislation, sources of contamination, environmental, and restriction measures for the implementation of parks.

### 3.1. Environmental Research—Contamination

#### 3.1.1. Jardim Primavera Municipal Park

In 2004, CETESB signaled to the Municipal Bureau of Green and Environment (SVMA) that only after a detailed investigation on site the environmental problems in the area could be identified and, from then on, determine recovery and mitigation measures to be adopted.

The SVMA, still in 2004, conducted an on-site inspection, verifying the existence of drains without maintenance, which would allow clogging and accumulation of gases. Furthermore, in view of the suspicion of contamination, the bureau considered it essential to carry out full studies to ascertain the necessary intervention measures [18].

The environmental study in Jardim Primavera municipal park area was executed in two phases. The first lasted from 29 March 2007 to 16 May 2007, and involved services and the geotechnical situation assessment studies, gas emanation investigations and confirmatory investigation (laboratory analyses of soil and groundwater). And the second phase was performed from 20 August 2007 to 1 April 2008, in which detailed research was undertaken, as well as preparation of the risk assessment and remediation proposals [20].

The inspections made did not detect conditions of instability, erosion, and leakage of surface percolates from the landfill, as well as exposed waste. It was also identified that the drainage and slurry treatment facilities were disabled.

Local geology was identified by a sequence of alluvial and floodplain deposition, resulting in stacking by sand packets, interspersed with the presence of thinner sediments such as silts and clay with varying colors [26].

The results obtained in step I indicated contamination in groundwater: metals (aluminum, barium, boron, total chromium, iron, manganese, nitrate); fecal coliforms resulting from the former effluent treatment plant (TEE) and total coliforms. And in the soil: methane gas present also in the drains and visiting boxes of SABESP arranged in the vicinity of the property. There were indications of the impact on the aquifer under study, especially in the downstream to the landfill in the direction of the Jacu stream by slurry [20].

In step II, aiming to expand the sampling and laboratory analyses, 36 investigative surveys were carried out, which required the installation of multilevel wells and monitoring wells. Soil contamination by metals (vanadium, iron, and aluminum) and groundwater by metals (aluminum, barium, cadmium, lead, cobalt, total chromium, iron, manganese, nickel)*,* fecal coliforms and total coliforms were identified.

In phase II, the presence of methane gas in the soil was also confirmed, concluding that there was an imminent risk resulting from concentrations higher than 2000 ppm. The exception was a grounded shallow well, with concentrations above 10,000 ppm in explosive conditions [20,22,26].

The results obtained from the analysis in stages I and II in 2008 indicated risks in the three different scenarios, their exposure routes and the receivers, Table 1.

In view of the results and the damage caused to the environment, the Public Ministry of the State of São Paulo (Ministério Público do Estado de São Paulo—MPSP), in 2012, filed a Public Civil Action in defense of the Environment under the terms of Law N.7347/1985 (regulates the Public Civil Action of Liability for Damage Caused to the Environment and provides other measures) against the City of São Paulo, requiring the total suspension of any work aimed at the implementation of the park [25].

The MPSP granted this injunction with the obligation of exhausting the gases present in the soil, if CETESB or GTAC did not find the absence of environmental risk, under penalty of a daily fine of ten thousand reais [27].ijerph-19-11933-t001_Table 1Table 1Results of Human Health Risk Assessment [28].ScenariosReceptorsExposure RoutesToxicological RiskCurrent (Deactivated Landfill)Commercial workers, security guards and construction workersAccidental ingestion, inhalation of particles and dermal contact with soil.AluminumFuture(Park)ChildrenAccidental soil ingestionAluminum and ironHypothetical(Park)Children, youth, adults, and the elderlyIngestion and dermal contact with groundwaterAluminum, Ironand Manganese


In 2013, a monitoring of possible gases existing in the area for a period of 2 years was proposed by SVMA. This monitoring served as an initial measure to update the current scenario, and to establish the procedures to be applied in the area, aiming at the health risks for employees and regulars of Parque Primavera [28].

Aiming at environmental compensation for damage, in the period from 29 January 2014 to 5 February 2014, phase I services were started, which included a soil *gas survey* and the installation of 40 gas monitoring wells.

Measurements of concentrations of volatile organic compounds and methane were performed in the underground facilities located around the study area (sewage, water, gas, drainage, and telephony networks), in addition to the inlet and outlet of pipes from existing buildings [21].

The results of the analyses obtained from volatile organic compounds in the soil showed concentrations above the standards adopted for benzene, tetrachloroethene, chlorobenzene, ethylbenzene, n-propylbenzene, 1,3,5-trimethylbenzene and 1,2,4-trimethylbenzene. Methane measurements performed in the three campaigns indicated concentrations above the Lower Explosive Limit (LEL), being: 22.5%, 20% and 7.5% above the LEL with explosive potential [29].

On February 2014, a soil sample was collected for the characterization and proper disposal of the residues, as established by ABNT NBR 10004/2004. The residue was characterized in excavated soil and classified as Inert IIB [21].

In phase II, gas levels were measured for a period of 2 years (2014 to 2016) with quarterly reports of their results, in addition to the intervention plan and the updated risk assessment [21] (Appendix A).

#### 3.1.2. Villa-Lobos State Park

Given the size of the area and the diversity of places and types of materials arranged in Villa-Lobos State Park, environmental investigations began to be carried out in two campaigns (7 May 2007 to 15 May 2007 and 24 July 2008 to 1 August 2008), considering the type of current occupation and the routes of exposure. Between 7 January 2008 and 18 January 2008, new samples were collected to confirm results for phthalates.

According to the surveys (from 0 to 15 m deep) carried out throughout the park, the presence of several materials at different depth levels was confirmed, as well as thickness and varied sizes, highlighting civil construction debris and dredging sediments of the Pinheiros River. However, the presence of plastic bags, pieces of dishes and glasses was also verified, indicating that the area may have received waste of domestic origin, but not on a regular basis. The presence of residues of industrial origin was not detected [19].

The analytical data of the collected samples indicated the existence of contamination in the subsurface soil (phenanthrene, PCBs, Bis (2-ethyl) phthalate and methane gas) and in groundwater (Indene (1,2,3-cd) pyrene, lead, and arsenic).

During the vapor surveys in the soil, an upward flow of gas under pressure was observed in one of the points investigated close to the administration buildings, with concentrations higher than 10,000 ppm or 100% UEL. The monitoring only indicated the presence of methane gas, generated from the degradation of organic matter. In view of this confirmation, the presence of gases in all underground utilities (rainwater galleries, passage boxes, drainage gallery) was monitored, which, according to CETESB emergency technical team, did not show significant explosiveness risk.

The exposure of the intake of free aquifer groundwater was not characterized in the area and its surroundings, due to installation of the wells in deeper parts. However, the existence of cisterns and other rainwater accumulation points used for irrigation was observed, characterizing the routes of dermal contact exposure and inhalation of water vapors for users and park employees, in open and closed environments.

With the need for more detailed research for soil vapors, the results of the 2009 analyses of groundwater samples showed that only zinc exceeded the intervention value (IV) established by CETESB (2005). It was found that although the zinc contaminant exceeded the IV, according to the risk assessment, the value found at the time was below the concentration that represents risk to the residential intake scenario for workers and users of the future park (consumption of 2 L/day).

Thus, it was recommended not to use local groundwater and monitoring every six-months, at least until the end of the investigation of the area. Continuing environmental management in the area, in November 2012, thirty areas were defined for surface soil sampling, twenty-nine monitoring wells for groundwater and for gas monitoring, fifty-nine for open environments and twenty areas for indoor environments, since it has few confined environments.

In view of the results obtained from the surveys, the identified residues were classified as non-hazardous and inert waste, and its use is allowed by the park itself. It was also verified the extensive presence of dredged material of the Pinheiros River, characterized by sandy clay, light gray to dark with odor and debris, deposited in several parts of the park [19]. 

In groundwater, contamination by metals (barium, boron, lead, manganese, nickel, and selenium) and volatile organic compounds (bromodichloromethane) and in the soil several points with phthalate and PAHS (polycyclic aromatic hydrocarbons) were identified: dietilexil phthalate, benzo(b)fluoranthene and benzo(k)fluoranthene, in addition to dissolved and total metals (cobalt and nickel).

For methane gas, residential use was considered, considering the use of the park by employees and users. Concentrations ranging from 52,300 ppm (5.23%) to 983,000 ppm (98.3%) were detected in the sampled monitoring wells, with a number above the flammability range of the compound, which is 5% to 15%. In the monitored indoor environments, no concentrations higher than 217 ppm (0.21%) were found [10]. However, considering the risk assessment, concentrations above the IV were detected for benzene and ethylbenzene in the wells located next to the buildings, which have underground environments, being the Orchid greenhouse and Reference Center in Environmental Education of the park.

In view of this situation, it was recommended to immediately install monitoring systems for volatile organic compounds—VOC—and methane in other constructions, in addition to ventilation, exhaust and/or inflation systems; continuous monitoring of indoor environments for the evaluation of intrusion; use of personal protective equipment (PPE) for any intervention made in the park and the maintenance of quarterly gas monitoring campaigns, evaluating VOC and methane (Appendix A).

### 3.2. Common Aspects—Case Studies

Considering that there is no specific Brazilian legislation addressing the subject, the legislation adopted as a reference for the process of requalification and the implementation of parks were those established by the environmental agency of the state: CETESB, according to similar ruling from the Environmental Protection Agency EPA of the United States of America, starting from 2005.

It is pointed out that the areas of those properties belong to the government, and that, in the mid-1960s, they were initially used for the activity of sand mining, followed later by the irregular disposal of urban solid waste and dredged materials, characterizing the two sites as dumps. Both sites were contaminated by PAHS, and methane in soil and water. It was evidenced that the main aspect considered for the requalification of the areas was the acceptable risk of exposure to human health, aiming at the safety of future users and the correct principle of operation of the parks.

In addition, in both cases there were similarities as to the measures required for the park area, such as the restriction of the use of groundwater for any purpose within the property; the final cover with clean soil preferably of clay origin; the restriction on the planting of fruit trees in the entire property; the construction of buildings in confined spaces and the monitoring of gases and groundwater.

### 3.3. Different Aspects—Case Studies

According to the contaminated area management procedure and CONAMA Resolution N. 420/2009, it was possible to identify that the objectives of the requalification of contaminated areas in São Paulo are not always achieved successfully and guaranteed by the Brazilian legislation.

There was unequal treatment in the requalification process, with emphasis on three specific aspects: the actors involved in the case, the role of civil society, and the action of the Public Prosecutor’s Office.

Regarding the actors involved, one can perceive the importance of private participation in projects of contaminated areas. In the area of the Villa-Lobos State Park, with the presence of the private sector, it was possible for its construction to begin quickly and with no major interruption during the environmental investigation, being fully completed as projected in 2013, demanding twenty-four years. As for the Jardim Primavera Municipal Park, it was observed that the municipal administration, as much as it has been committed to meeting all the requirements for safety and requalification of the area, was unable to have an active and swift participation in the process.

Another aspect observed in the Villa-Lobos State Park was the participation of the neighborhood and resident population of the surrounding area. This participation of the population occurred from the deactivation of the area until its total conversion into a park. Civil society, through non-governmental organizations (NGOs), actively participated in the entire process of requalification of the area, given its benefits from future land use and occupation.

On the other hand, regarding Jardim Primavera Municipal Park, the participation of the community occurred mainly through complaints and claims made to the environmental agency, denouncing the risks that the former controlled landfill represented, and the problems encountered in the attempt to build the park.

Another issue of great relevance identified in the requalification process was the participation of the Public Prosecutor’s Office of the State of São Paulo.

In Brazil, according to the Constitution of the Federative Republic of Brazil (art. 225, §3) and Law 6.938/1981 (art. 4, item VII) of the National Environmental Policy, it was established that once the environmental damage has been caused, it must always be fully repaired [22]. This total repair is considered a determining economic aspect in the sphere of sustainability because it adheres to the Polluter-Pays Principle, which emerged in principle 16 of the Declaration of the United Nations Conference of RIO 92, defined as: “the one who infects must, in principle, bear the costs of contamination” [23].

However, when analyzing this idea in a contaminated area such as the studied parks, there are divergences between the current norms related to the recovery of contaminated areas versus repair of environmental damage. These divergences, although they are still under debate in major technical events, often lead to legal conflicts between the actors involved, which can directly compromise the urban requalification and sustainable development of land use in important regions of the city of São Paulo.

As in the areas of the parks there is buried waste, it is recorded that it is not feasible to fully recover the environmental damage caused, that is, it is impossible to have an ecologically balanced environment as recommended by the entire legislative framework focused on environmental damage, and its recovery.

Another point that is worth mentioning is that the Villa-Lobos State Park, because it was able to quickly start the first environmental studies to investigate contamination in the area, was able to avoid the legal process and the deviation from the planned purposes for the area, such as that identified by the Jardim Primavera municipal park area. In a way, it is pointed out that the lawsuit promoted by the population in this site ended up contributing to the State, showing the importance of the principle of participation in the decision-making process.

In general, the importance of effective efforts and investments in the institutions and individuals involved can be highlighted. One solution that could be applied in these cases is knowledge management or 3D creativity management. 

This management is based on a conceptual structure in three large blocks to obtain something innovative, being: 1-block: of absorption and filtering of information; 2-block: processing of creativity and the 3-block: results of innovation [30]. In the case of Jardim Primavera Park, it is evident that the government has failed to absorb and filter the information to monitor, control and manage the entire process of requalification of the area to develop appropriate responses and strategies [31]. In the Villa-Lobos State Park, with civil society commitment and investments, in a public–private partnership, it was possible to process and collect all information effectively and efficiently throughout the requalification process. Later, it was possible to evaluate, compare and monitor the entire system to develop viable strategies and to complete the park, achieving an innovative result for the city of São Paulo.

## 4. Discussion

In the city of São Paulo, enactment and care of green areas and public parks may be the responsibility of the municipal or state administration, which is the responsibility of SVMA through DEPAVE, or SIMA.

In the contaminated areas of the case studies, it was evidenced that the practice of sand extraction was very common at the time and many areas that were converted into green areas, previously housed extraction pits. These pits, for the most part, were used for disposal of urban solid waste without adequate engineering recommendations and resulted in soil contamination and risks to human health. The current situation has no relation to the original uses. As examples, other parks, such as the Embu Das Artes Ecological Park, the Olympic Rowing Lane of USP, Ibirapuera Park and Toronto City Park, have been transformed into public areas, taking advantage of the mining pits [32].

Urban parks worldwide are considered places of integration and exercise of citizenship for all social classes and ages, as well as ideal for the development of permanent programs and campaigns of environmental education. Once inserted in urban areas, they promote various benefits to the population, the environment, and the conservation of biodiversity [33,34].

Studies confirm the benefits and importance of parks in the socioeconomic, public health and environmental spheres. These include improving air quality, soil permeabilization and environmental comfort; increasing physical, social, and psychological health; promoting well-being and health; preventing chronic diseases such as depression; social inclusion; and improving mood and self-esteem, evidencing that contact with nature leads to positive results for health in the short and long term [14,35,36].

Studies found that just five minutes of walking in green areas brings improvements in people’s mental health, such as mood and self-esteem. Such evidence suggests that sedentary people and/or people with mental problems would have mental health benefits by committing to short-term exercises in accessible green areas [35]. Thus, parks are places that promote the feeling of well-being of users, sports practices, greater socialization, and stimulation of the identity of the community with the place, playing a motivator and social inclusion role. 

Additionally, a reduction of 20–25% of diseases is estimated among children who do not live near contaminated places; the increase in the value of residential property between 19–24%; the creation of jobs and the increase in local taxes for new uses and functions, evidencing the importance of a healthy, clean, and accessible environment for the population [36].

Considering that in both areas there was soil and groundwater contamination, it might be stated that environmental investigations were managed adequately, following the technical protocol of the environmental agency CETESB.

As for the different aspects identified, the importance of the participation of investors and construction companies is emphasized, as they assume two roles, that of financial-economic viability and the execution of the revitalization itself. In addition, they seek and involve other actors concerned, and together they promote the necessary synergy for the requalification of the area [37]. In this scenario, the difficulty of the municipal administration of the Jardim Primavera Municipal Park of having to exercise the role of the government and that of investor at the same time can be justified, making it difficult to complete the project, which lasts inconclusive until now.

Thus, a possible solution would be that of reviewing the planned project for the area and verifying whether it is possible to implement another use for the benefit of the community, such as the project carried out at the municipal landfill in Brick Township, New Jersey, USA. That landfill received materials from contaminated sewage and liquids, which led to groundwater contamination and was added in 1983 to the EPA’s list of national priorities for its cleanup [38]. Due to the extent of the contamination, the EPA required Brick Township to install a waterproof cover in the landfill to prevent rainwater from infiltrating the landfill floor. In addition, it required that the municipality conduct a long-term monitoring groundwater and use of drinking water for human consumption.

Therefore, Brick Township decided to turn this former landfill into a solar power facility, providing electricity to the surrounding government buildings, residences and community parks, Figure 8.

Another important situation is the direct insertion of the population even before the elaboration of the project, as in the case of Villa-Lobos Park, where it ended up stimulating environmental education, not only to inform about the problem, but to provide sufficient and adequate knowledge for better understanding, transparency, and decision-making regarding the problem [38]. In addition, with the creation of the Orientation Council of the Villa-Lobos State Park, the State was able to arouse community interest, later counting on its support, dialogue and dissemination of all actions concerning the requalification process of the area, in a receptive and transparent way.

Social participation is one of the main ways to stimulate the population in identifying, planning, and implementing actions that help create healthier environments and improve quality of life. However, in the decision-making process regarding Jardim Primavera Municipal Park there was not a participatory moment of direct and conscious exchange through education, but only meetings and consultation spaces without participatory instance between representatives and represented. 

In other words, the right to information is the one that establishes a duty-power, that is, the whole of the public power and the community (State and Society) in a participatory and active way of the different groups and segments interested in the formulation and execution of the decision-making circumstances [28].

The public authorities must assume and present a more educational, effective, and qualified function in the implementation of the Jardim Primavera Municipal Park regarding the real problems that affect the area. It is also necessary to objectively provide access to information, and better disseminate and explain it to the population, showing how this requalification will be made for the region, clearly addressing all their interfaces.

A successful case that highlights the importance of civil society in environmental requalification participation and direct government participation is Chambers Gully Park in Australia. Chambers Gully Park, located in the suburbs of Adelaide, Australia, was a local landfill.

The recovery of the area was carried out through the joint action of volunteers and residents of the area and with the help of government funds. The area today is a sanctuary for wildlife, including mainly kangaroos and koalas that are often seen perched on the park’s eucalyptus trees (Figure 9), as well as a 9.6 km trail with a 3-h duration. The circuit cannot be accessed directly by road, but it is an easy 1.2 km walk to reach an access point, from Chambers Gully parking lot [39].

As far as environmental liabilities are concerned, it was evident that the Public Prosecutor’s Office needs to reach consensus with the public authorities so that it can admit, at first, another form of environmental repair other than integral.

According to CONAMA Resolution No. 420/2009, the process of recovery of contaminated areas does not guarantee the promotion of environmental quality prior to contamination, as it aims to adopt corrective measures based on the set of actions aimed at isolating, containing, minimizing, or eliminating contamination, enabling them to be recovered for a use compatible with the established goals, thus adopting the principle of declared use. With this rehabilitation of the area, a new occupation is possible, whether residential, commercial, or agricultural, a procedure adopted worldwide.

A solution would be public policies aimed at environmental management and recovery based on mechanisms of tax incentives and specific laws that stimulate the requalification of contaminated areas. Among the options we can mention the national funds for orphan contaminated areas, list of priority contaminated areas, creation of taxes, partnership between agencies of different spheres, participation of civil society in the process of requalification, decentralization and flexibilization of soil protection legislation and financial support and partnership with the private sector, of great importance to achieve a successful requalification case.

The present work directly contributes to a more sustainable planning for cities since it considers a viable alternative for the use and occupation of soil in urban space.

Although one of the case studies—Parque Jardim Primavera—has not been successfully completed, Villa-Lobos Park has evidenced that this practice can be carried out safely, as well as in the international context. This sustainable tool aims at government efforts, funding, public policies, and public–private partnership, thereby promoting the environmental quality of life of cities and population, reducing crime, irregular occupation, degraded places and preserving biodiversity.

## 5. Conclusions

The studied cases allowed us to know that the areas contaminated by solid waste, considered as successful cases in the city of São Paulo, had the direct support of civil society and the public authorities complemented with the private sector. Another issue is that for these urban projects to be completed with greater depth and ease, it is necessary to reformulate and/or modify the Brazilian legislation, especially in the thesis of Integral Repair advocated by the Public Ministry versus Management of Contaminated Areas.

It was evident that before sanctioning a law creating a park, the government needs to stimulate/insert direct private participation in the projects of contaminated areas, because they are essential for unfinished cases such as the Jardim Primavera Municipal Park. It should also actively and directly insert civil society support in the requalification process, enabling practical, in loco, knowledge. In addition, it needs to seek the support of other competent bodies such as urban policy, public health, tourism, aiming to create better solutions and projects for the requalification of waste-contaminated areas as a tool for sustainable urban development of land use and promotion of quality of life.

For future actions in other areas contaminated by municipal solid waste, not only for the city of São Paulo but for the world, it is first recommended to study the contaminated area/site in a thorough manner, so that it is possible to identify its interaction with the environment and its surroundings. In addition, in requalification planning it should be identified whether the ideal is really a park project for the area, and whether there is another utility option.

Another important factor is to respect the natural conditions of the place, changing only what is necessary, and when altered, seek projects that require low maintenance costs, thus avoiding new degraded spaces, abandoned and/or idle by the city. It is necessary to consider the gains and impacts for the region with the new requalification of the area, as well as its attributions/responsibilities, whether social, environmental, economic, or cultural.

The research’s positive points include the scope and interdisciplinary view on the subject. Finally, it is recommended to further explore the theme discussed in the research and the evaluation of the actions of the public authorities in relation to areas contaminated by waste. Although actions aimed at the creation of parks in these areas have been proposed, it is important that there is a strengthening of legislation, planning, supervision and monitoring so that success cases are obtained as a result.

## Figures and Tables

**Figure 1 ijerph-19-11933-f001:**
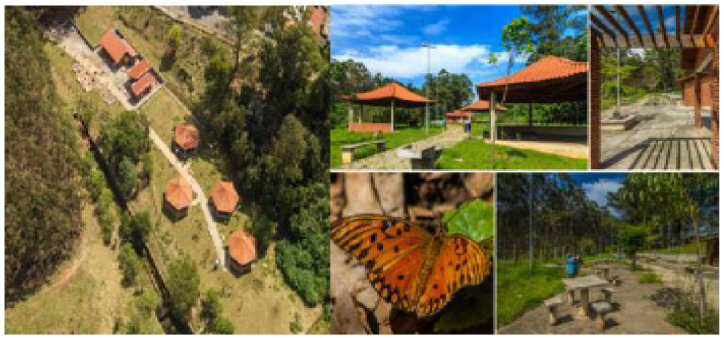
Municipal Park Jardim Primavera [18].

**Figure 2 ijerph-19-11933-f002:**
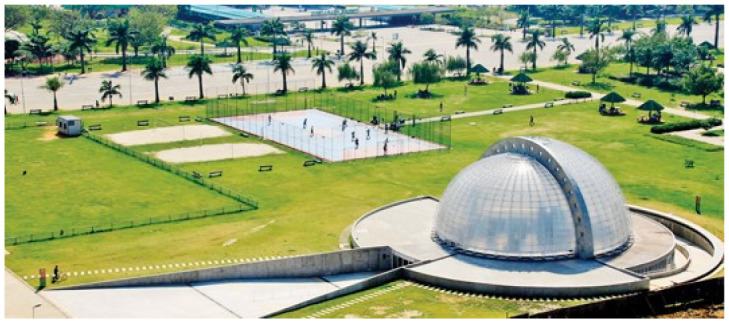
Villa-Lobos State Park [19].

**Figure 3 ijerph-19-11933-f003:**
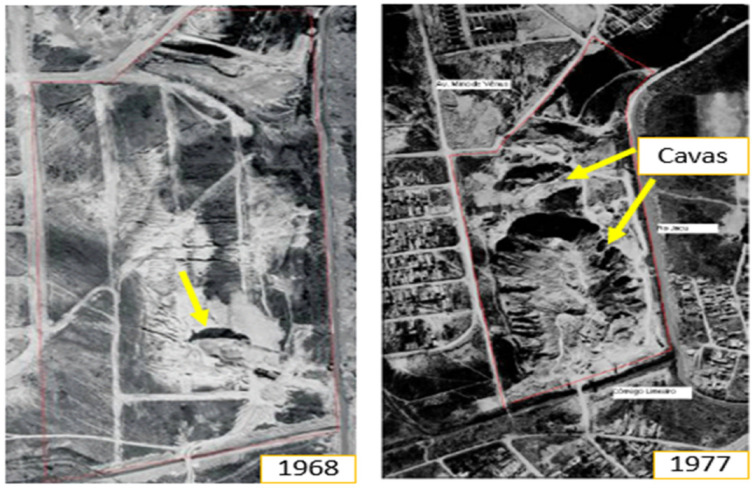
Pit formation in 1968 and 1977. Municipal Park Jardim Primavera [18].

**Figure 4 ijerph-19-11933-f004:**
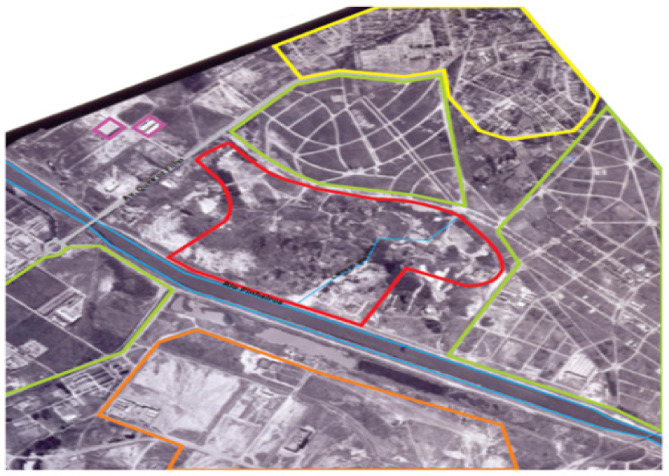
Villa-Lobos site. Occupation of the area and neighborhood in 1958. Colors: Red: Villa-Lobos Park Area; Brown: Earth movement; Green: green area/allotment; Blue: Body of water; Yellow: residences; Purple: industries; Orange: University of São Paulo [19].

**Figure 5 ijerph-19-11933-f005:**
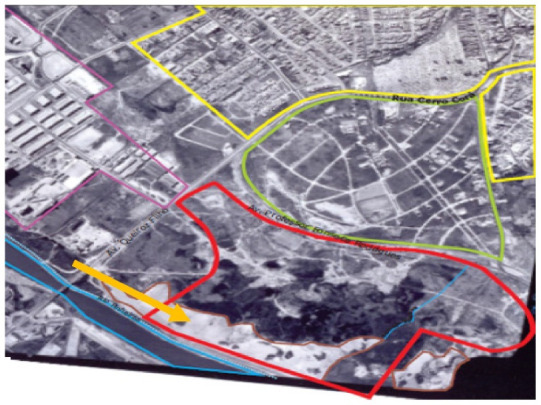
Villa-Lobos site. Earth movement in the south part of the area in 1968 [19].

**Figure 6 ijerph-19-11933-f006:**
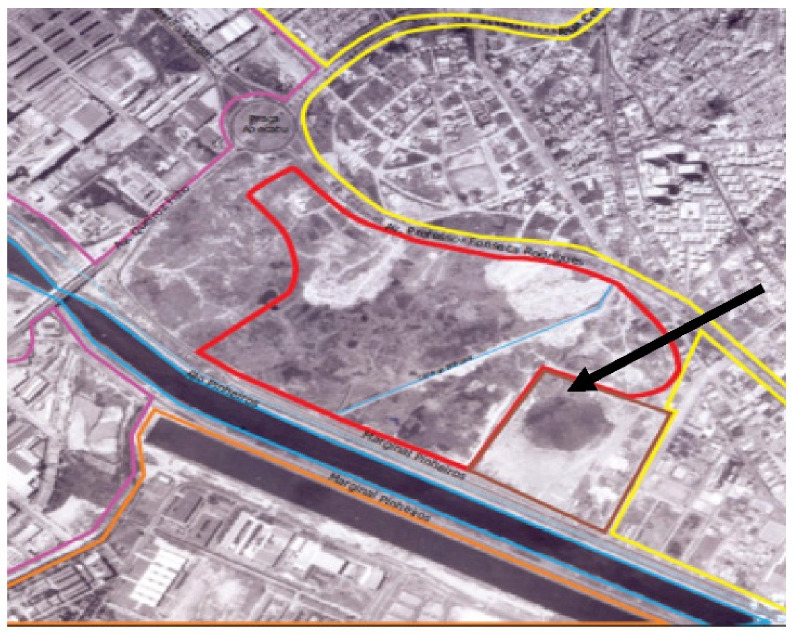
Villa-Lobos. Disposal of dredged material next to the old sand mine in 1977 [19].

**Figure 7 ijerph-19-11933-f007:**
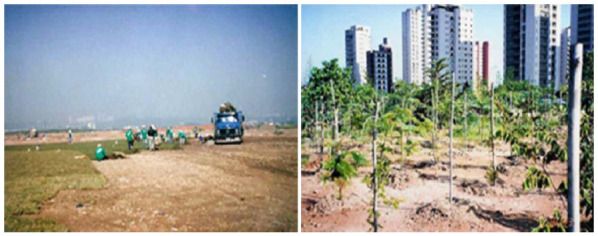
Building of Villa-Lobos State Park in 1989 [24].

**Figure 8 ijerph-19-11933-f008:**
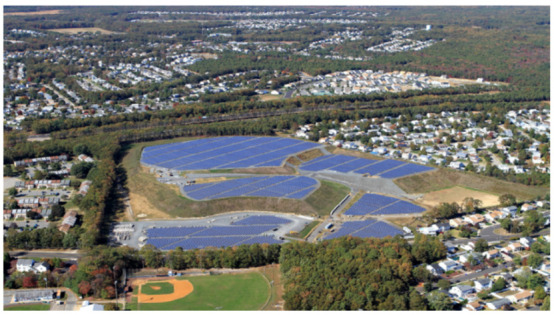
Brick Town landfill. Solar panel view [38].

**Figure 9 ijerph-19-11933-f009:**
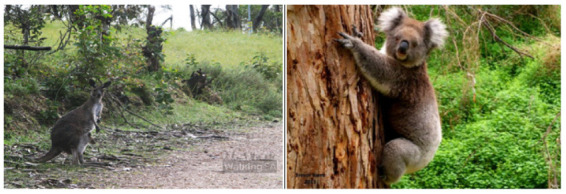
Chambers Gully Park, Adelaide, Australia [39].

## Data Availability

Not applicable.

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
