# Peer review of "Potential Transformation of Contaminated Areas into Public Parks: Evidence from São Paulo, Brazil"

_ijerph, 2022, doi:10.3390/ijerph191911933_

Round 1
Reviewer 1 Report
The authors of the reviewed article deal with an important and up-to-date topic.
The article is well structured - it contains all the elements required for scientific texts. The proportions between the particular sections are well balanced. The authors try to present the re-use of degraded brownfields as urban green areas in the context of improving public health. Unfortunately, this thread was dealt with only marginally and rather vague in the introduction to the article and again in the summary/conclusion. The authors focused primarily on the process of converting contaminated wastelands into parks in Sao Paulo. However, only two such cases are covered in detail, one of which has not been fully implemented. Thus, the material for comparative analyses and studies is rather scarce. What is more, the article shows that only one of the described sites was a waste dump site and, consequently, environmental hazards occurred in its vicinity.
The question therefore arises whether converting contaminated sites into urban parks is indeed an effective method of solving environmental problems in cities.
However, even with such limited comparative material, it was possible to draw conclusions and attempt recommendations / recommendations for further projects of this type implemented in Brazilian cities. Unfortunately, the authors have limited themselves to basic comparisons and the conclusions are timid.
There are no evident factual errors in the text. Only in the description of works related to the establishment of the Jardim Primavera park, there is information that phase II of environmental research (including comprehensive investigations) ended earlier than the phase I (p. 8, lines 326-331). It is not known, however, whether this is a methodological error committed in reality (in this case it should be emphasized in the conclusions) or whether it is a mistake made by the authors at the stage of processing the source material.
The article has illustrative deficiencies - in particular there is a lack of a figure showing the aerial photo of the second case study with the context (similar to the figure 3) or any other visual meeting this purpose.
Although I am not a native speaker, I do notice some flaws in the use of the English language. Therefore, a proofreading and linguistic correction are necessary.
Author Response
The introduction was improved with inclusion of data on contaminated sites in the world and other Brazilian cities.
The results were also improved with inclusion of a table as complementary material, describing the rehabilitation processes in both parks, making easy the comparison of the cases.
Beneficial health effects of green areas and parks were presented in light of new literature included in the article. Examples of transformation of other contaminated/dump areas into parks in São Paulo were mentioned even though the article focuses in the 2 case studies which were studied in depth.
Regarding the point that " the description of works related to the establishment of the Jardim Primavera park, there is information that phase II of environmental research (including comprehensive investigations) ended earlier than the phase I (p. 8, lines 326-331)". The information was corrected. Sorry for this typing error.
To attend the suggestion that "the article has illustrative deficiencies - in particular there is a lack of a figure showing the aerial photo of the second case study with the context (similar to the figure 3) or any other visual meeting this purpose". - aerial photos of the second case were included in the article.
Tha manuscript was further revised by a native English speaker.
Reviewer 2 Report
Reviewer’s comments:
I have comments on your paper titled “Transformation of Contaminated Areas into Public Parks”. Your paper topic is interesting and your work can potentially contribute to the literature on environment management, urban management, and public health. Yet, substantial revisions are required to improve your work to be considered published by the journal.
Specific comments:
1. The paper title should be revised a bit. For example, you can rename the title: “Potential transformation of Contaminated Areas into Public Parks: Evidence from XXX”.
2. Your presentation of the results of the paper is not clear. This section must be reorganized to make it more clear and more logical for readers to follow. For example, Section 3.1. should be moved/placed before the Results Section, which you can name “Background”. It is also possible to create a clear diagram that depicts the events associated with the stakeholders involved over time. Alternatively, you could have a table with key project information such as time scale, stakeholders, outcomes, good/bad things, and so on for each case.
3. The Discussion Section seems still weak. You stated that social participation and civil society play an important role in triggering and transforming contaminated areas into parks, but you didn't go into detail. For your discussion, you should also consider the costs, public-private partnerships for environmental projects, and the importance of communication and/or collaborations among stakeholders for project success (Q. H. et al. Vuong, 2022; Q. H. Vuong & Napier, 2014).
4. The author(s) also should elaborate on the theoretical and practical contributions of your study.
5. There are some key references that you may consider using for your revising.
Vuong, Q. H. et al. (2022). Covid-19 vaccines production and societal immunization under the serendipity-mindsponge-3D knowledge management theory and conceptual framework. Humanities and Social Sciences Communications, 9(1), 1–12. https://doi.org/10.1057/s41599-022-01034-6
Vuong, Q. H., & Napier, N. K. (2014). Making creativity: the value of multiple filters in the innovation process. International Journal of Transitions and Innovation Systems, 3(4). https://doi.org/10.1504/IJTIS.2014.068306
Author Response
- The article was renamed, as suggested.
- Results were better organized. Item 3.1 History was changed to before results and the name changed to Background, as suggested. A table with all the important dates and actions for both parks was made and included as complementary material, as asked.
- The discussion section was improved with more detailed information regarding the rehabilitation processes and inclusion of new references, as recommended
- The discussion was improved regarding practical and theoretical aspects.
- One of the 2 references suggested was included.
Round 2
Reviewer 1 Report
no further comments
Reviewer 2 Report
Thank you for carefully addressing my comments. The revised manuscript is now much improved, and I have no further comments on your work.